



# Mapping methane plumes at very high spatial resolution with the WorldView-3 satellite

Elena Sánchez-García[1*], Javier Gorroño[1*], Itziar Irakulis-Loitxate[1], Daniel J. Varon[2], Luis Guanter[1]

[1]Research Institute of Water and Environmental Engineering (IIAMA), Universitat Politècnica de València, Valencia, Spain
[2]School of Engineering and Applied Sciences, Harvard University, Cambridge, MA, USA
* These authors contributed equally to this work.

*Correspondence to*: Elena Sánchez-García (elsncgar@upv.es)

**Abstract.** The detection of methane emissions from industrial activities has been identified as an effective climate change mitigation strategy. These industrial emissions, such as from oil and gas (O&G) extraction and coal mining, typically occur
as large plumes of highly concentrated gas. Different satellite missions have recently shown potential to map such methane plumes from space. In this work, we report on the great potential of the WorldView-3 (WV-3) satellite mission for methane mapping. This relies on its unique very high spatial resolution (up to 3.7 m) data in the shortwave infrared part of the spectrum, which is complemented by a good spectral sampling of the methane absorption feature at 2300 nm and a high signal to noise ratio. The proposed retrieval methodology is based on the calculation of methane concentration enhancements
from pixel-wise estimates of methane transmittance at WV-3 SWIR band 7 (2235-2285 nm), which is positioned at a highly-sensitive methane absorption region. A sensitivity analysis based on end-to-end simulations has helped to understand retrieval errors and detection limits. The results have shown the good performance of WV-3 for methane mapping, especially over bright and homogeneous areas. The potential of WV-3 for methane mapping has been further tested with real data, which has led to the detection of 26 independent point emissions over different methane hotspot regions such as the O&G
extraction fields in Algeria and Turkmenistan, and the Shanxi coal mining region in China. In particular, the detection of very small leaks (<100 kg/h) from oil pipelines in Turkmenistan shows the game-changing potential of WV-3 to map industrial methane emissions from space.

## 1 Introduction

Widespread awareness of the accelerated increase of methane and other polluting gases in the atmosphere has heightened the
need for new technologies to rapidly identify and control emission sources. During the last decades, increasing methane levels have been related to the rise of global warming risk. Reducing or stabilizing emissions would allow for a prompt decrease of atmospheric concentrations thanks to methane's relatively short lifetime of about 9 years (Saunois et al., 2020). Therefore, controlling the main contributors of methane emissions is seen as an effective option to mitigate climate change. Methane emissions from the fossil fuel industry represent a critical opportunity for mitigation, given the large number of
uncontrolled emission point sources in oil and gas (O&G) and coal production areas worldwide (Jackson et al., 2020).



Substantial advances have been made in the last years towards the detection and quantification of methane point emissions from space (Jacob et al., 2016). The GHGSat satellite constellation was the first remote sensing system optimized for the monitoring of methane point emissions (Varon et al., 2018) and is allowing the detection of a large number of anthropogenic methane emissions around the world as shown in Varon et al. (2019, 2020). More recently, imaging spectrometers covering

the entire 400-2500 nm region (known as hyperspectral imagers) have also shown their potential for mapping methane plumes from space such as the ZY1 AHSI, Gaofen-5 (GF5) AHSI, and PRISMA missions discussed in Irakulis-Loitxate et al. (2021a; 2021b) and Guanter et al. (2021). GHGSat and these hyperspectral missions share a relatively high sensitivity to methane concentration enhancements and a spatial sampling in the range of 25-50 m, but their acquisitions are sparse in time and space. The Sentinel-2 (S2) multispectral mission, originally developed for land applications, can help to overcome this

limitation with frequent and spatially-continuous observations (Varon et al., 2021), albeit with a substantially lower sensitivity to methane. Landsat-7 and 8 missions, whose spectral channel configuration can provide a methane detection sensitivity comparable to S2 (see Fig. 1), offer the possibility for enlarging timeseries (Landsat-7 has operated since 2003). However, Landsat has coarser spatial and temporal resolution (30 m vs. 20 m, and 15 days vs. 5 days) compared to S2. Methane mapping efforts with all those missions greatly benefit from the synergistic use of the TROPOspheric Monitoring

Instrument (TROPOMI) onboard the Sentinel-5 Precursor satellite (Hu et al., 2018; Gouw et al., 2020; Lauvaux et al., 2021). TROPOMI data allow to identify regions with strong methane concentration enhancements (Sadavarte et al., 2021) where the aforementioned missions can focus on the detection of single emitters.

The present work describes a new breakthrough in this quickly developing field towards the global mapping of methane point emissions. Using very high-resolution shortwave infrared (SWIR) measurements from the WorldView-3 (WV-3)

satellite, we have mapped methane plumes from different locations around the Earth at a spatial resolution of 3.7 m (E. S. Imaging, 2020). Positive plume detections have been obtained from O&G extraction fields in Algeria and Turkmenistan, and the Shanxi coal mining region in China. Until now, methane retrievals at such high-resolution were only possible for airborne instruments, like the AVIRIS and AVIRIS-NG spectrometers operated by NASA JPL (Frankenberg et al., 2016; Jongaramrungruang et al., 2019; Cusworth et al., 2019; Duren et al., 2019). The WV-3 instrument combines a high spatial

resolution with a high signal-to-noise ratio (SNR) and rich spectral coverage of the strong methane absorption feature around 2300 nm (see Fig. 1). Furthermore, the onboard system includes pointing capabilities that are able to deliver a daily revisit or better over critical infrastructure. These unique features allow WV-3 to fill an important observational gap in international satellite methane monitoring capabilities.



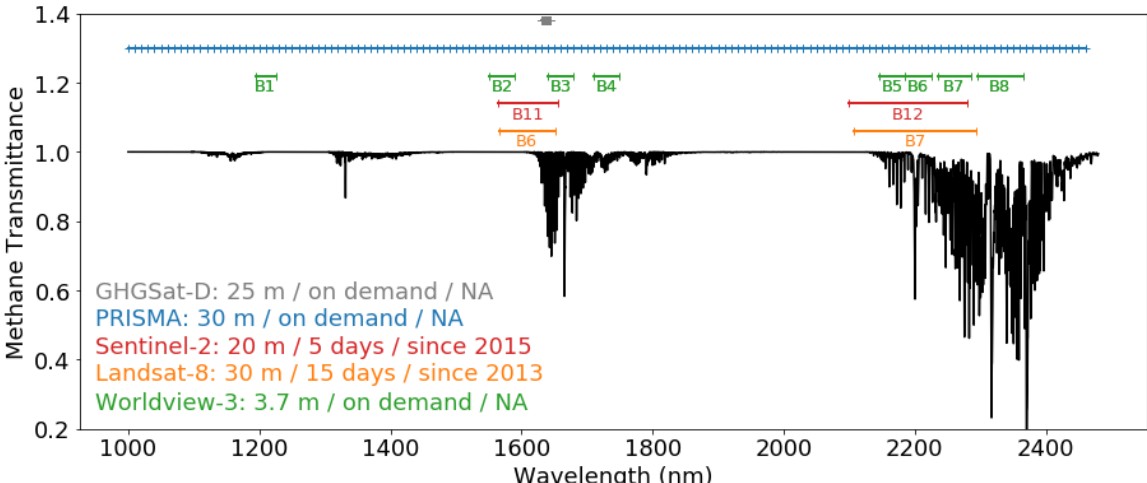

**Figure 1. Comparison of the spectral sampling in shortwave infrared of different spaceborne instruments with potential for methane mapping. All of these instruments sample the methane absorption feature around 2300 nm. The spatial and temporal sampling characteristics of each mission (spatial resolution/temporal resolution/temporal coverage) are also provided for each mission.**

Mapping methane emissions at this very high spatial resolution enables a substantial improvement of emission detection thresholds. Accurate identification of the particular infrastructure elements responsible for the emissions and more precise quantification of emission rates is also made possible by the unprecedented spatial resolution of WV-3.

## 2 Materials and Methods

### 2.1 Methane retrieval

Methane retrieval algorithms for hyperspectral instruments typically estimate methane concentration enhancement by fitting highly-resolved observations in the SWIR spectral region to a modelled radiance spectrum (Thorpe et al., 2014; Jacob et al., 2016). In a single overpass, hyperspectral instruments onboard satellites or airborne missions can resolve the SWIR spectral region with a typical spectral resolution ranging from 0.1 to 15 nm.

For multispectral instruments like S2, Landsat, and WV-3, the entire SWIR spectral region cannot be resolved. Nonetheless, it is possible to retrieve methane concentration enhancements from increases in methane transmittance within the image. To calculate methane transmittance, we normalize the radiance at a spectral channel affected by methane absorption by a "methane-free" reference band, i.e. with no excess methane present. In the absence of multi-temporal data, this reference can be built from one or several neighboring channels mostly "methane-free". WV-3 contains a richer spectral design with 8 SWIR bands at 3.7 m spatial resolution (see Fig. 1) and narrower bandwidth as compared to the S2 mission. Specifically, it contains two bands (B7 and B8) in regions of strong methane absorption and spectrally close bands (B5 and B6) at less than 100 nm separation (as opposed to the 600 nm spectral distance between S2 B11 and B12 bands). WV-3 images in the SWIR



are processed with a Time-Delayed-Integration (TDI) of 16 lines. This contributes to a superior SNR as compared to other multispectral missions such as S2.

Thus, the proposed rationale to detect methane plumes is based on an estimation of the plume transmittance defined by the ratio of a methane-sensitive band such as WV-3 B7 or B8 with excess methane present, and the equivalent named "methane-
free" reference band. Mathematically the plume transmittance $T_{plume}$ can be described as:

$$T_{plume}(\lambda) \sim \frac{L}{L_{ref}} = e^{-AMF \cdot \sigma_{CH_4} \cdot \Delta XCH_4} \tag{1}$$

where L and $L_{ref}$ represent the radiance of the methane-sensitive band and the "methane-free" reference band respectively. AMF refers to the air mass factor, $\sigma_{CH_4}$ (ppm$^{-1}$) refers to the methane absorption cross section, and $\Delta XCH_4$ (ppm) refers to the methane concentration enhancement. The latter expresses the increment produced by the plume from the background
methane present in the atmospheric column.

We have tested that both the spectral response function (SRF) knowledge and water vapor absorption constitute minor sources of uncertainty in the estimation of $T_{plume}$. The sensitivity study setup a reference scene with a simulated methane plume (Q=1000 kg/h). In order to test the impact of the SRF knowledge on the methane retrieval, the SRF central wavelength has been shifted for the convolution of the methane transmission. A shift of ±5 nm has been selected as a
"typical" tolerance of a multispectral mission design. The results show that the impact of spectral shifts on B8 is lower than 1% and larger errors were observed in B7 (up to 8%) due to the larger changes in methane absorption within this band. The water vapor column has been modified in a range from 0 to 40 mm representative of extreme values in the atmosphere. The results for water vapor errors can reach up to 5% for B8 and just 1% for B7.

The approach to obtain the $L_{ref}$ "methane-free" reference band (L in the absence of excess methane) is based on the idea that
this could be approximated using other mostly "methane-free" spectral channels. The simplest form of this model would be the ratio between B7 and B5 which are spectrally close and correlated (see Fig. 1).

From the several options that have been considered (see Fig. 6), the selected method for the estimation of this "methane-free" band is based on a multiple linear regression (MLR) of B1-B6 with B7 as the target band (B7/B7$_{MLR}$ method). Despite B8 being more sensitive to methane absorption, this band has shown lower spectral correlation with B1-B6 bands as
compared to B7 (see results in subsection 3.1). Out of the six bands considered for the regression, two of them (B3 and B6) are marginally sensitive to methane. In the case of B3, its weight on the regression is small and has negligible impact thus it has been decided not to include this band. However, in the case of B6, its spectral closeness to B7 improves the regression and outweighs the impact that the residual methane sensitivity of B6 produces on the retrieval. Several tests have shown that the integrated mass enhancement (IME) lost could be around the 7%.





The methane plume quantification with WV-3 is obtained by isolating the methane enhancement in Eq. (1). Then, $\Delta XCH_4$ becomes a function of the AMF and the radiance ratio $L/L_{ref}$ of SWIR B7 (2235-2285 nm). A look-up table (LUT) can be generated so that the relationship between these two quantities and the methane enhancement is established and expressed as follows:

$$\Delta XCH_4 = \frac{-log(L/L_{ref})}{AMF \cdot \sigma_{CH_4}} \qquad (2)$$

Figure 2 shows the LUT relationship between the methane plume transmittance and the methane column enhancement ($\Delta XCH_4$) for different values of AMF. The figure also highlights the optimized spectral design of WV-3 with respect to other multispectral missions such as S2. It illustrates how both WV-3 B7 and B8 cover a wider range of plume transmittance values as compared to S2 band 12. Furthermore, it shows how the sensitivity increases with the AMF values although this is at the expense of potential shadowing and surface directional effects (Guanter et al., 2021).

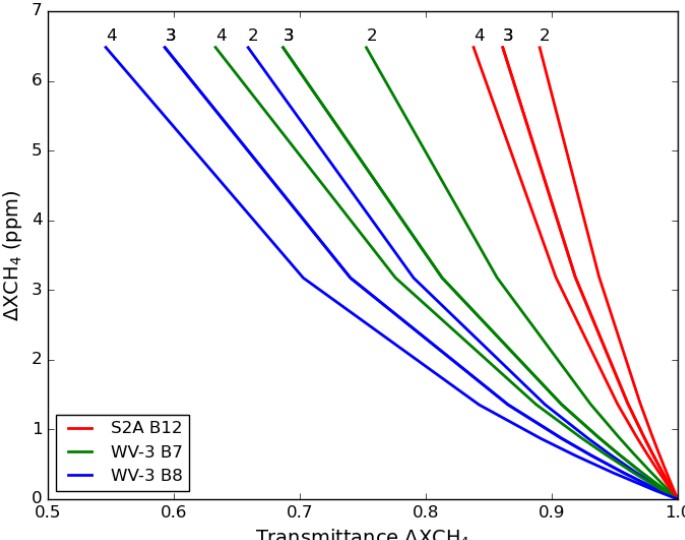


**Figure 2. Relationship between the methane plume transmittance values and the methane column enhancement ($\Delta XCH_4$) for the methane absorption bands (~2300 nm) of S2A B12, WV-3 B7 and WV-3 B8. The curves are given for three different values of AMF representing different angular conditions ranging from 0 to 60 degrees (labelled above as 2 to 4).**

**2.2 Detection of methane plumes**

The detection of methane plumes begins with the visual inspection of the retrieved $\Delta XCH_4$ maps. It is generally easy to distinguish the plumes from the background in the retrieved maps due to the low random noise and the characteristic shape of the plumes. This noise is generated by the inherent instrument noise, misregistration between the bands and several surface features that confuse the retrieval by having a similar spectral signature to the methane. Once the plumes have been

initially identified, it is checked that their shape is consistent with the Goddard Earth Observing System-Fast Processing (GEOS-FP) wind speed and direction data. That is, the shape of a gas plume is expected to originate at high $\Delta XCH_4$ values



and decrease progressively downwind. Once the plumes have been verified, they are collocated with high spatial resolution true color images of the area in order to identify the underlying infrastructure responsible for this emission. For this process, we use Google Earth images and the first SWIR band (B1~1200 nm) of the WV-3 images.

The next step of the detection processing zooms in on each real plume to isolate it from the background in order to compute the total area covered by the emission. The methodology proposed in Thompson et al. (2016) is based on a statistical significance of a rectangular region of 500 samples directly upwind of the plume meanwhile Varon et al. (2018) uses the statistical significance ($p \leq 0.05$) associated with a 5×5 window Student's t-test. In both cases a median filter is set to smooth the image and remove outliers.

Here, a semi-automatic process has been implemented where the plume is isolated by applying a mask at the 95% confidence level, and a square dilation mask of several pixels a-posteriori. Similar to the median filter, a mask dilation ensures that plume tails are included in the detected area. Moreover, it guarantees a plume selection with a reduced number of discontinuities as expected from the spatial distribution of a gas plume. Finally, the detected outliers in the vicinity of the plumes are removed through feature recognition. Techniques such as clustering properties and blob detection measures that

allow to get a kind of image classification and remove those features that are not part of the emission but present high $\Delta XCH_4$ values in the retrieval. Sometimes the radiance of the satellite bands is also used in this discrimination process.

## 2.3 Estimating emission flux rates

Once the plumes are correctly defined, the associated flux rate Q is obtained by applying the so-called IME model which

refers to the measure of the total excess mass of observed methane widely explained in Frankenberg et al. (2016) and Varon et al. (2018). The $\Delta XCH_4$ values are converted from ppm to ppm m units and to kg (Thompson et al., 2016; Duren et al., 2019) by considering the 3.7 m of WV-3 pixel spatial resolution, the Avogadro's law where 1 mole of gas occupies 22.4 L, the estimation of 0.01604 kg per mole of methane, and a factor of 8000 that accounts for the column height assuming an atmosphere of 8 km. Then, we compute IME in kg for each plume as the total sum of the $\Delta XCH_4$ pixels enclosed by the

plume mask. The last step relates the emission flux rate with the calculated IME as:

$$Q \left(\frac{kg}{h}\right) = \frac{U_{eff} \left(\frac{m}{s}\right) \cdot IME \text{ (kg)} \cdot 3600}{L \text{ (m)}} \qquad (3)$$

where L is a plume length scale in m (square root of the entire area covered by the plume pixels), and $U_{eff}$ the effective wind speed derived from Weather and Research Forecasting Model large-eddy simulation (WRF-LES) according to Varon et al. (2018) methodology.

For this study, the WRF-LES simulations have been tuned at the spatial resolution and measurement uncertainty of WV-3 (~4 m resolution and 1-sigma white noise varying from 120 to 370 ppb depending on the site; see Fig. 8). Then, six IME





calibrations have been performed for each of these three 1-sigma noise levels and considering two different emission ranges: $Q \in [100, 3000]$ kg/h and $Q \in [100, 500]$ kg/h. Figure 3 shows the fits obtained for the extreme cases out of the six IME calibrations which correspond respectively to $\sigma = 120$ ppb and $Q \in [100, 3000]$ kg/h (the highest signal case), and to $\sigma = 370$

ppb and $Q \in [100, 500]$ kg/h (the lowest signal case).

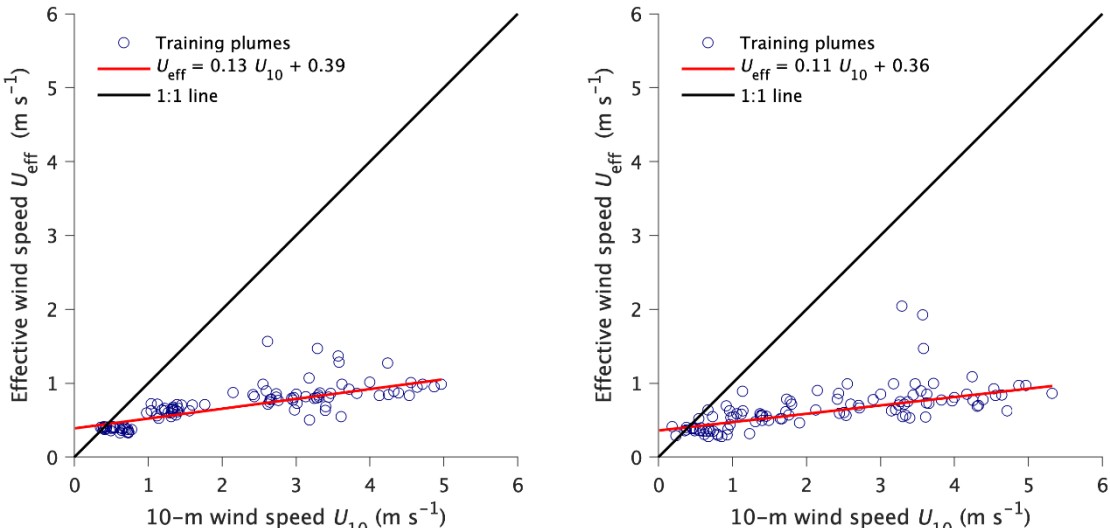

**Figure 3.** Scatter plots relating the effective and local 10-m wind speeds ($U_{eff}$ vs $U_{10}$) in the IME method, characterized with WRF-LES training plumes. The graph on the left shows the fit obtained for $\sigma = 120$ ppb and $Q \in [100, 3000]$ kg/h. The graph on the right shows the fit for $\sigma = 370$ ppb and $Q \in [100, 500]$ kg/h. Black and red lines show respectively the 1:1 fit and the linear

regression fit.

The similarity achieved between the six individual fits leads to the adoption of the following formula that considers the middle values of the coefficients obtained from the calibrations:

$$U_{eff} \left(\frac{m}{s}\right) = 0.12 \cdot U_{10} + 0.38 \tag{4}$$

In this proxy, the measurable 1-h average 10-m wind speed ($U_{10}$) derives from the two north-south and east-west wind components of GEOS-FP dataset (https://portal.nccs.nasa.gov/datashare/gmao/geos-fp/das/) at the satellite acquisition time and for the location of each plume. Consider that GEOS-FP is a meteorological reanalysis product with a resolution of $0.25° \times 0.3125°$ (Molod et al., 2012).

This calibration has been set for small plumes because the LES model domain is only ~200 m across. Then, this would be

expected not to be accurate for larger plumes causing underestimates of Q because these plumes, even sampled at 4-m resolution, are seeing transport over a much larger part of the boundary layer than the small ones. Therefore, the $U_{eff}$ calibration set for PRISMA in Guanter et al. (2021) would be more properly used for those plumes over 200 m in length.



For each Q value, we have estimated an associated uncertainty similarly to the method used in Cusworth et al. (2020), Guanter et al. (2021), and Irakulis-Loitxate et al. (2021a; 2021b). The uncertainty formalism is based on a Monte Carlo

propagation (JCGM, 2008) of the uncertainties in IME, $U_{10}$, and the coefficients of Eq. (4) through Eqs. (3-4). Here, the uncertainty of the input $U_{10}$ is set to a conservative 50% uncertainty $k=1$[1] estimate in GEOS-FP $U_{10}$ data consistent with the ~1.5-m/s error standard deviation in wind speed given by Varon et al., (2020). The uncertainty in the IME is calculated based on the uncertainty combination of different pixels with an associated "retrieval noise". This is the consequence of instrument noise, misregistration, and surface heterogeneity. Finally, an uncertainty $k=1$ of 0.01 for both the slope and

intercept of Eq. (4) is included in the Monte Carlo propagation. This is a proxy of the regression sensitivity to different scenarios such as different flux rate ranges or noise levels previously discussed.

The different error samples in the Monte Carlo propagation have been treated as uncorrelated. This can be justified because there is no expected relationship between the errors in IME, $U_{10}$ and the intercept/slope in Eq. (4) as they are largely obtained from independent processes. A weak correlation of ~0.28 between the slope and intercept errors in Eq. (4) has been

empirically determined but considered of negligible impact on the overall budget. The result of the Monte Carlo propagation is an error distribution associated with the calculated Q value. The standard deviation of this error distribution is considered the uncertainty $k=1$ for the Q of each plume under the assumption of a normal distribution. Both the accuracy of the IME model and its relationship with the length scale L of the plume are considered to be out of the scope of this work.

## 2.4 Sensitivity analysis based on end-to-end simulations

The potential of WV-3 for methane mapping has been assessed through a sensitivity analysis based on end-to-end simulations. The simulations are generated by convolving WV-3 TOA radiance scenes with synthetic methane plumes generated with the WRF-LES.

The WRF-LES simulations are set at 50 m × 50 m spatial resolution, 100 W m$^{-2}$ sensible heat flux and a mean wind speed of 3.5 m s$^{-1}$ as in Cusworth et al. (2019). These original plumes are upsampled to 3.7 m in order to match the spatial resolution

of WV-3. The plumes are further scaled from an initial source rate $Q$ of 100 kg/h to the desired flux. The upsampling and scaling of the reference plume has no effect on the sensitivity analysis but a small interpolation effect that generates a second-order error. An example of the resulting methane plume is shown in Fig. 4 with $Q$ of 1000 kg/h.

---

[1] The coverage factor, $k$, is a factor applied to the combined standard uncertainty that specifies the fraction of the probability distribution that the uncertainty represents (JCGM, 2008). For $k=1$ and an ideal normal distribution, it is equivalent to the standard deviation of the probability distribution.


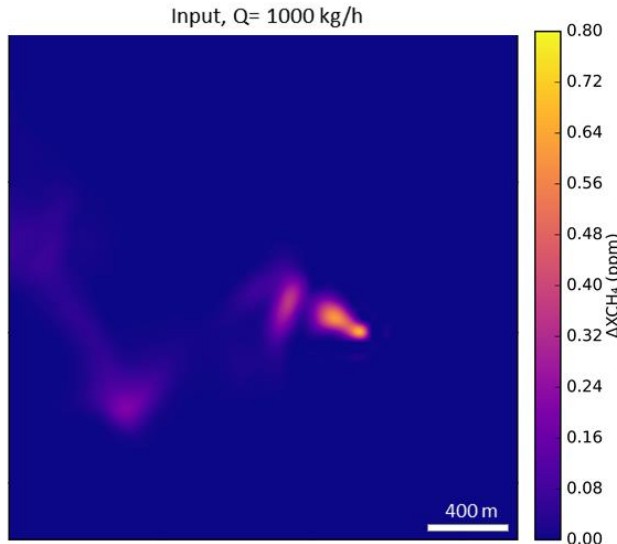

**Figure 4. Example of plume with Q 1000 kg/h used as a reference over the same subset of 2.5 x 2.5 km shown in the bottom row of Fig. 5, Fig. 6 and Fig. 8.**

Similar to Cusworth et al. (2019), the spectral optical depth of the methane plume $\tau(\lambda)$ is calculated as the sum of 72 layers of the atmosphere. At each layer, we multiply the HITRAN absorption cross sections ($\sigma_H$; Kochanov et al., 2016), the vertical column density of dry air (VCD), and the methane volume mixing ratio enhancement ($\Delta$VMR). We obtain a methane plume transmittance map at a fine spectral resolution by applying Beer's law. Finally, this fine spectral resolution map is convolved with the WV-3 SRF and the result is a methane plume transmittance map for each one of the WV-3 spectral bands.

The calculated plume transmittance for each band is multiplied with the TOA radiance scene. The resulting simulation incorporates a real scene from WV-3 meaning that effects such as instrument noise or angular pixel dependencies are already included in the simulation. The a-posteriori convolution of the band radiance with the plume transmittance produces a small bias that is partially compensated.

The end-to-end simulations have been generated for three different sites in Algeria, Turkmenistan and China whose spectral response behave differently regarding their surface conditions (see Fig. 5). The WV-3 images used for the simulations have been acquired on the 29[th] of December 2020 in Algeria, the 27[th] of April 2021 in China, and the 29[th] of March 2021 in Turkmenistan. Fig 5. displays in the top row an RGB snapshot of the three sites. The area defined by the red square on the RGB snapshots shows the WV-3 SWIR B7 subsets displayed at the bottom row of the figure which are the ones used for the end-to-end simulations.



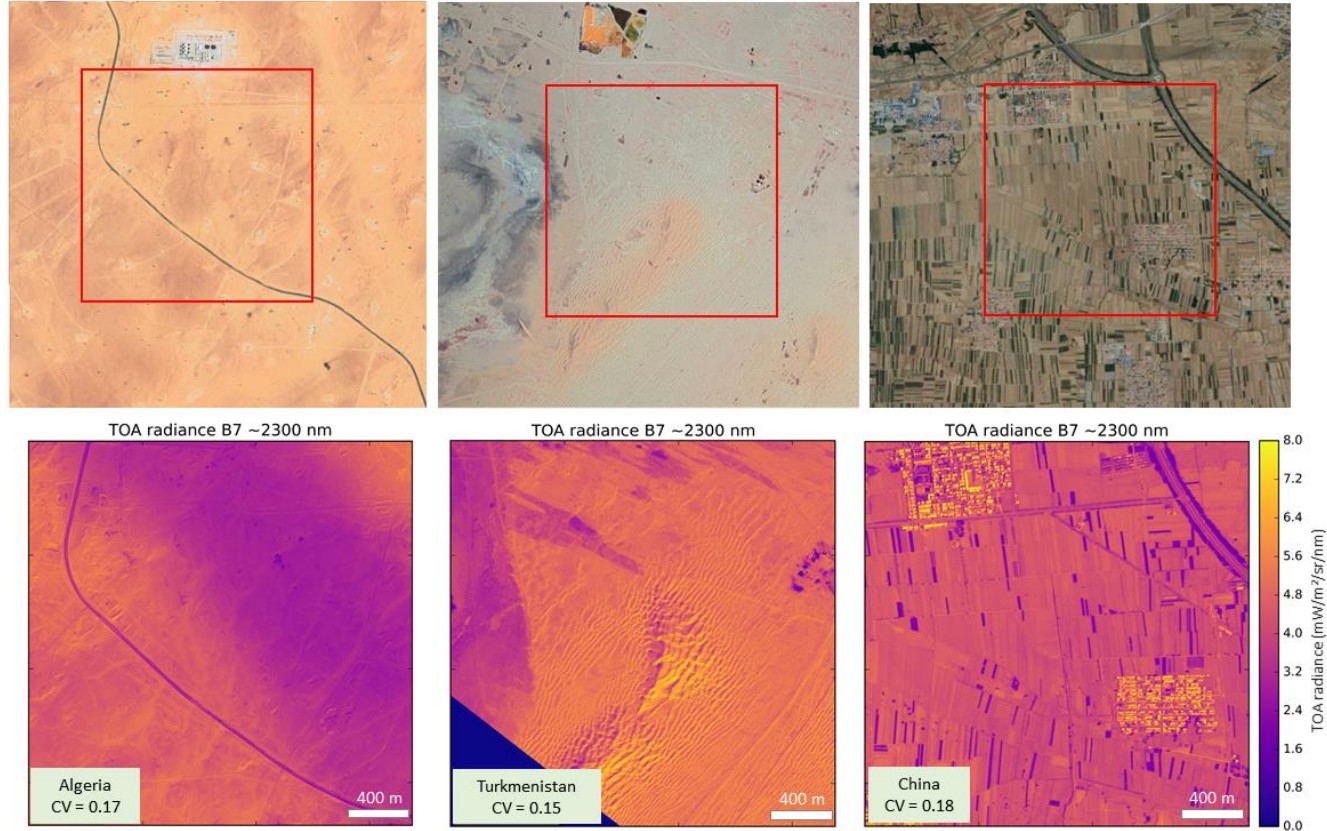

**Figure 5. Sites considered for end-to-end simulations. Top panels represent the sites in RGB color from © Google Earth imagery (at earlier dates). The areas defined by a red square of 2.5 x 2.5 km (676 x 676 pixels) are the WV-3 SWIR subsets used as input**
230  **for the sensitivity analysis and where simulated plumes have been added. The © WV-3 SWIR B7 subsets are displayed at the bottom row and the values of the Coefficient of Variation (CV; standard deviation over mean) are labelled for each site.**

Among the chosen sites, the one in Turkmenistan shows the lowest spatial variations in the SWIR B7, whereas the one in China is expected to be the most challenging site for methane plume detection due to its higher spectral and spatial heterogeneity. The Algeria site covers a largely homogeneous area similarly to Turkmenistan although it contains some
235  cirrus/cloud shadows at the right half of the selected subset that alter the radiance in SWIR B7.

Finally, it is important to remark that the sites selected for the simulations are realistic insofar as they are well-known areas of O&G and coal production facilities with known methane emissions (Irakulis-Loitxate et al., 2021a; 2021b; Varon et al., 2021; Cusworth et al., 2021).





## 3 Results

### 3.1 Retrieval performance with simulated plumes

The end-to-end WV-3 simulations defined in subsection 2.4 have been used here to test the performance of different methane retrievals. Figure 6 presents the $\Delta XCH_4$ maps at two sites that result from the application of different approaches to calculate a "methane-free" reference band. The retrieval using a radiance ratio of B8 against the MLR of B1-B6 with B8 as the target band shows the largest sensitivity as compared to the other two retrievals using B7. However, the result also exhibits a pronounced impact of outliers (e.g. the dunes in Turkmenistan or the roads in Algeria) due to the larger spectral distance from B8 and B1-B6. The results using a radiance ratio of B7 against the MLR of B1-B6 and that of a single ratio of B7 against B5 present similar results. Nonetheless, the former results in a clear shape of the plume and a slight decrease of outliers. Thus, the calculation of the plume transmittance as a radiance ratio of B7 against the MLR of B1-B6, described in subsection 2.1, is selected as the best compromise between detection sensitivity and low outlier generation.

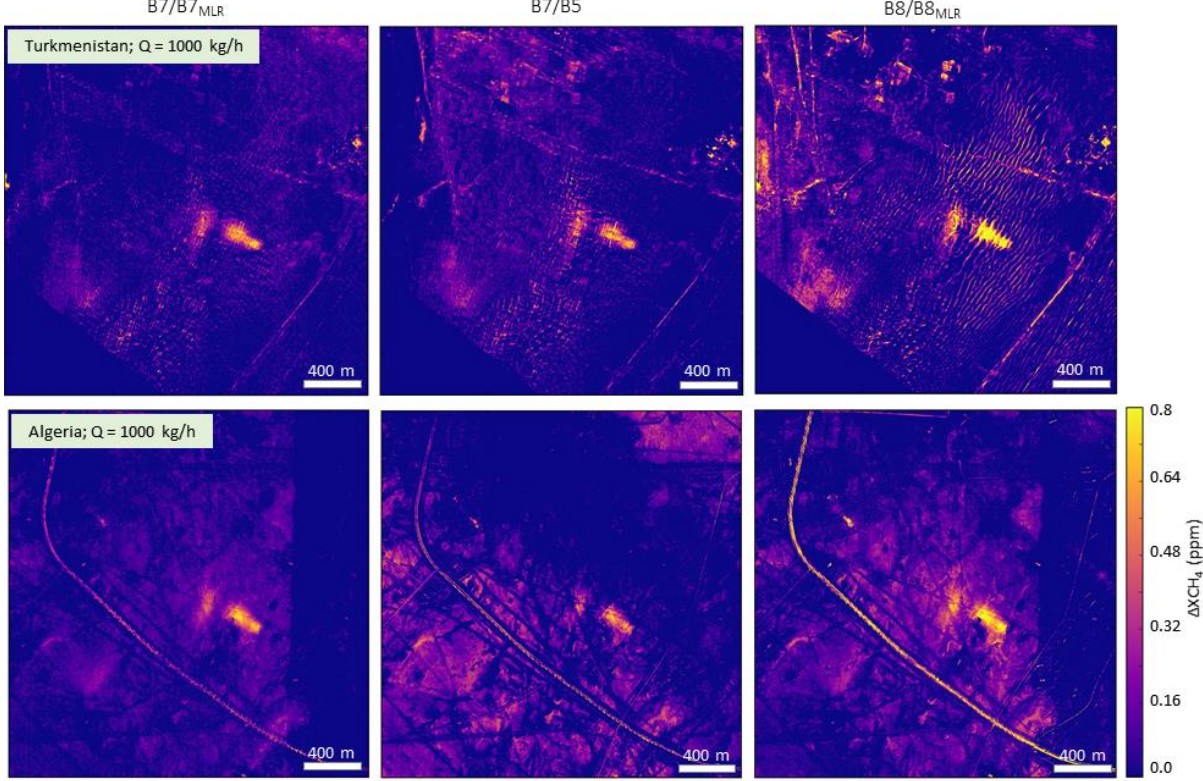

**Figure 6. Comparison between different $\Delta XCH_4$ retrieved maps for the simulations at two sites (with the input plume of Q=1000 kg/h in Fig. 4), and obtained by carrying out three different retrievals: from left to right: the ratio between B7 and B7 predicted (MLR with B1 to B6); the ratio between B7 and B5; and the ratio between B8 and B8 predicted (MLR with B1 to B6).**

The selected methodology has been performed with the end-to-end simulations at various sites and for different Q levels. The method takes the radiance ratio $B7/B7_{MLR}$ which defines a proxy of the methane plume transmittance that can be



associated with a methane enhancement value according to Beer's law. Different flux rates have been simulated leading to a set of $\Delta XCH_4$ maps that have been further processed in order to isolate the plume from the background as explained in subsection 2.2.

Figure 7 displays the retrieved $\Delta XCH_4$ maps for an input flux rate of 500 and 2000 kg/h at the three sites of study. A simple glance at the enhancement maps suggests that the best results are obtained in the Turkmenistan and Algeria sites. The Algeria maps show a discontinuity on the right side of the image. This effect is caused during the WV-3 product processing and could be due to focal plane discontinuities or other acquisition artefacts. Based on the results presented here this effect must be considered a minor one largely due to the spectral correlation nature of the error. Nonetheless, this effect is compensated in the results presented in subsection 3.2. Finally, the enhancement maps of China present many features that

alter the retrieval and require higher methane plume signals in order to positively detect them.

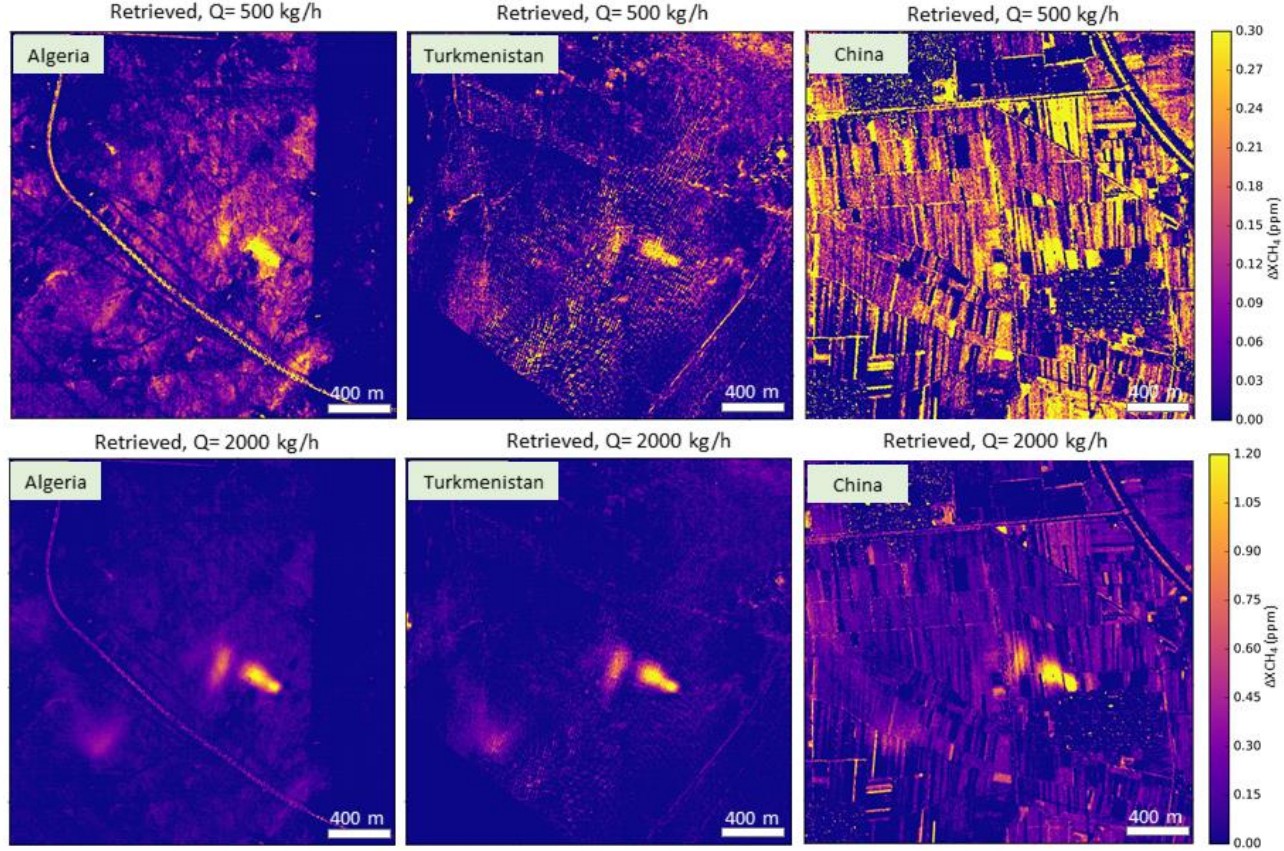

**Figure 7. $\Delta XCH_4$ maps for the simulations at the three different sites analyzed (same subsets as Figs. 5 and 6) and two values of emission flux rate Q 500 kg/h and Q 2000 kg/h.**

The performance of the retrieval is easier to understand if we attend to the background noise of the $\Delta XCH_4$ maps. Figure 8

shows the distribution of the retrieved $\Delta XCH_4$ maps for the three sites presented in subsection 2.4. The data represents a





subset of 600x600 pixels of the original images. The histograms indicate the variation of the noise levels across the different sites from a low level of 120 ppb in Algeria up to 370 ppb in China (1-sigma retrieval errors used to set Eq. (4)). These values, as expected, are sensibly larger than other missions with a lower spatial resolution due to the increased surface variability. For example, the retrieval noise reported by Guanter et al. (2021) using PRISMA data is almost half of the values

shown here for similar sites. Another observed difference is the non-normal nature of the retrieval noise in WV-3 (the same plots for the retrieval noise in PRISMA show an almost perfect Gaussian fit). Further investigation has concluded that this non-normal behavior is not produced by any step in the retrieval itself but is present in the radiance ratio $L/L_{ref}$. Different ratios were investigated all producing a non-normal distribution and it remains even after applying a spatial binning to e.g. 30 m. Thus, it is likely that this non-normal behaviour is the consequence of a noise pattern between B7 and the rest of the

bands (e.g. due to the large number of TDI stages which might enhance the prevalence of non-normal effects). However, it requires a better understanding of the calibration and instrument specifications that are not currently available.

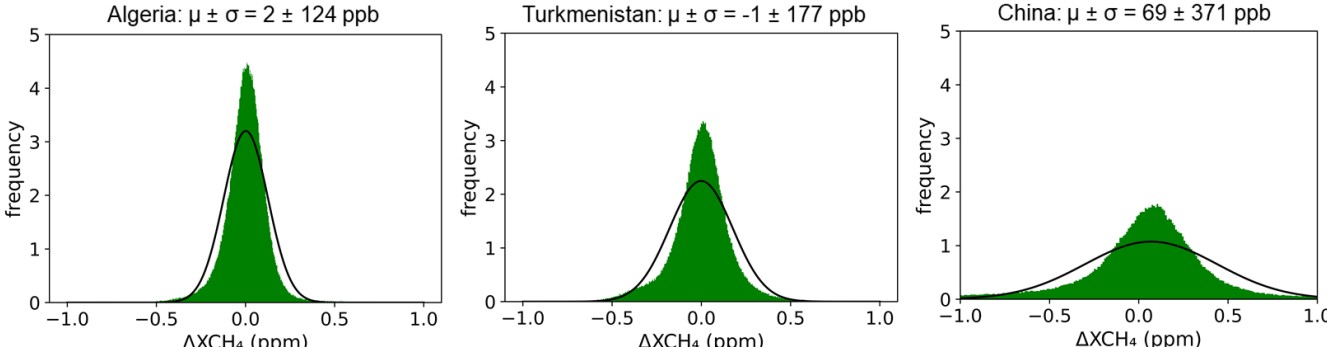

**Figure 8. Histograms of the resulting 600x600 $\Delta$XCH₄ subset areas using three original WV-3 images with a different type of surface. The mean and standard deviation values of the distributions for each site are shown in the chart titles.**

The methane plumes presented in Figure 7 have been isolated from the background considering the noise levels in Figure 8 and the procedure explained in subsection 2.2. Figure 9 displays the relationship between the retrieved IME values and the known reference input for the simulations at the three sites. The simulations in China and Turkmenistan have been performed at a Q step size of 500 kg/h whereas in the case of Algeria the step size is reduced to 100 kg/h. The results highlight a general underestimation of the retrieved IME values. For example, considering an input flux rate of 1000 kg/h,

the results show an IME underestimation of 11% for Algeria rising up to 56% for the case of China. The missing fractions of the total IME are larger at all input levels for the China case and the lowest fraction is associated with the Algeria simulations. This is a similar result to other studies such as Guanter et al. (2021) and comes to reinforce the importance of surface heterogeneity on the performance retrieval. It is important to remark that the IME loss scales in absolute values with the value of simulated Q. However, in relative terms this loss tends to become smaller.

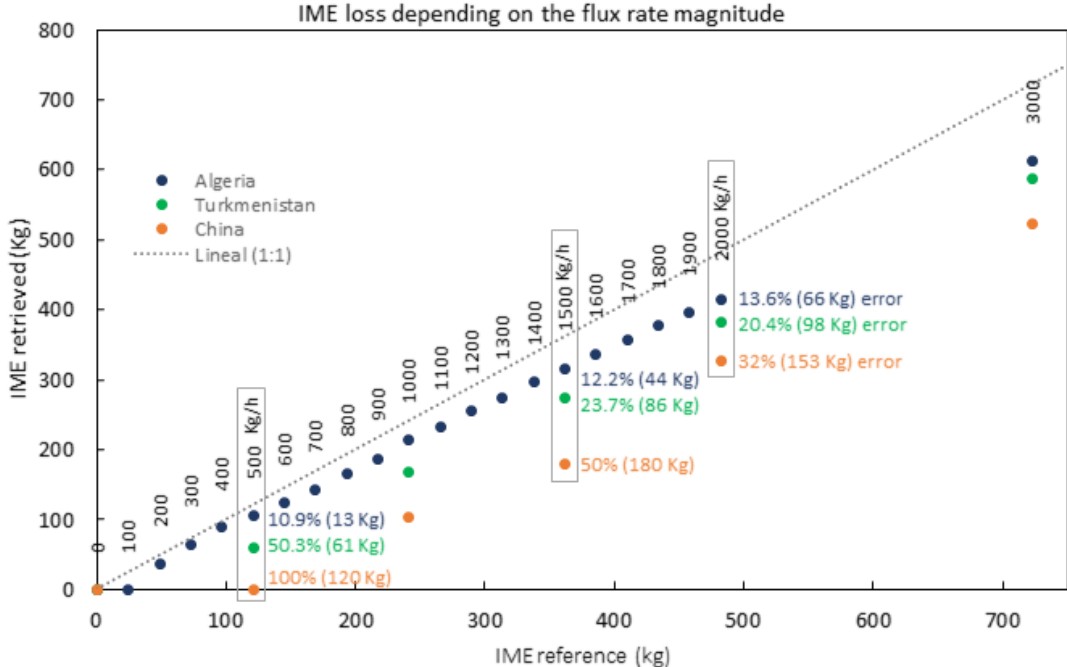


**Figure 9. Test of methane detection limits. The scatter plot relates the different referenced IME values -according to each input flux rate Q labelled above the dots- and the reconstructed IME from the retrieved ΔXCH4 maps of Algeria (blue dots), Turkmenistan (green dots), and China (orange dots) sites. For Q 500 kg/h, Q 1500 kg/h and Q 2000 kg/h, the IME losses are labelled in colors for each site showing the % of error set in the IME reconstruction, and the total underestimated magnitude of**
**the plume in Kg. Black line represents the 1:1 fit.**

### 3.2 Real case studies of methane detection

Different ΔXCH4 maps are presented here derived from real WV-3 images. We work with a total of seven SWIR WV-3 images covering four different areas in Algeria, Turkmenistan North and Turkmenistan South (both methane hotspot regions located near the shores of the Caspian Sea, in western Turkmenistan), and China. In the case of Algeria, two images of 25
km$^2$ area each belonging to the Hassi Messaoud Oil field have been processed, the eastern one was acquired on the 29[th] December 2020, and the western one was acquired on the 17[th] January 2021. The image in Turkmenistan North corresponds to the Goturdepe O&G extraction field where two images of 50 km$^2$ each have been acquired on the 10[th] April 2021 (Eastern area) and on the 29[th] March 2021 (Western area). The Turkmenistan South site (near the mud volcanoes area) contains the Korpezhe O&G field where two more images of 50 km$^2$ each have been acquired on the 29[th] March 2021 for both eastern
and western areas. Finally, we present an image of the Shanxi coal mines in China (west of the Zhangze reservoir) that covers an area of 25 km$^2$ and was tasked for acquisition on the 27[th] of April 2021.

All of these areas correspond to known regional methane hotspots where TROPOMI commonly detects substantial methane concentration enhancements (Lauvaux et al., 2021) but without succeeding in detecting the sources due to their moderate resolution.



Our results with WV-3 reveal the precise locations of 26 different point source emissions (listed in Table 1) over the total area of 275 km² analyzed. Accurate coordinates of each point source are provided thanks to the high spatial resolution of WV-3 SWIR images and, in combination with the Google Earth imagery (<2.5 m/pix), that provides enough information to determine the underlying infrastructure responsible for the emission. From the calculation of the IME, we derived the Q values estimated for all plumes as presented in subsection 2.3. The plumes presented here range from the 30 kg/h emission

flux rate and 50 m length plume detected in Turkmenistan North (Western image) to the massive emission flux rate of 35000 kg/h extending up 2 km in Turkmenistan North (Eastern image).

**Table 1. List with the disaggregate information for the 26 methane plume locations found, here grouped in 21 estimated Q flux rate. By order: "Site" refers to the name of the production field, "Emitter location" refers to the coordinates of the detected emitting points, "Date" is the image acquisition day, "n°pix" is the number of pixels enclosed by the plume mask, "L" the longitude of the plume, "$u_{10}$" and "$u_{eff}$" the local 10-m and the effective wind speeds, "IME" the total sum of the methane concentration pixels, and "Q" the estimated emission flux rate for each single or grouped plume.**

| Site | Emitter location | Date | n°pix | L (m) | $u_{10}$ (m/s) | $u_{eff}$ (m/s) | IME (kg) | Q (kg/h) |
|---|---|---|---|---|---|---|---|---|
| Hassi Messaoud Oil field (Algeria) | Pipeline (31.778°N, 5.995°E) | 2020/12/29 | 3363 | 215 | 6.14 | 2.53 | 74 | 3100 ± 1300 |
| | Pipeline (31.768°N, 6.000°E) | | 3155 | 208 | | | 57 | 2500 ± 1000 |
| | Pipeline (31.797°N, 6.011°E) | | 924 | 112 | | 1.12 | 13 | 500 ± 200 |
| | Pipeline (31.742°N, 5.895°E) | 2021/01/17 | 1846 | 159 | 2.37 | 0.66 | 41 | 600 ± 100 |
| Korpezhe O&G field (Turkmenistan South) | Ground Flare (38.494°N, 54.198°E) | 2021/03/29 | 16956 | 482 | 3.93 | 1.78 | 1353 | 13000 ± 4800* |
| | Ground Flare (38.557°N, 54.200°E) | | 13188 | 425 | | | 287 | 3100 ± 1100* |
| Goturdepe field (Turkmenistan North); Eastern image | Pipeline (39.474°N, 53.743°E) | 2021/04/10 | 20786 | 533 | 9.63 | 3.71 | 1390 | 35000 ± 15000 |
| | Pipeline (39.462°N, 53.775°E) | | 7424 | 319 | | | 118 | 5000 ± 2200 |
| Goturdepe field (Turkmenistan North); Western image | Two-point emitters (39.498°N, 53.636°E & 39.497°N 53.638°E) | 2021/03/29 | 44689 | 782 | 1.84 | 1.07 | 496 | 2400 ± 700 |
| | Four plumes around (39.485°N, 53.663°E) | | 8900 | 349 | | | 105 | 1200 ± 300 |
| | Pipeline (39.480°N, 53.671°E) | | 3957 | 233 | | | 11 | 200 ± 60 |
| | Two small plumes around (39.480°N, 53.671°E) | | 185 | 50 | | | 0.8 | 30 ± 10 |
| | | | 198 | 52 | | | 0.9 | 40 ± 20 |
| | Five smaller southern plumes around (39.469°N, 53.649°E) | | 116 | 40 | | 0.6 | 3 | 200 ± 30 |
| | | | 105 | 38 | | | 3 | 200 ± 30 |
| | | | 537 | 86 | | | 14 | 400 ± 70 |



| | | | | | | | |
|---|---|---|---|---|---|---|---|
| | | | 1842 | 159 | | 61 | 800 ± 200 |
| | | | 1020 | 118 | | 27 | 500 ± 90 |
| Coal mines in Shanxi (China) | Xiligaocun (36.257°N, 112.923°E) | 2021/04/27 | 6470 | 298 | 5.89 | 129 | 3800 ± 1600 |
| | Wangzhuang Beili two-point emitters (36.247°N, 112.989°E & 36.246°N, 112.989°E). | | 7042 | 310 | | 1.09 | 126 | 3600 ± 1500 |
| | Taoyuancun (36.234°N, 112.946°E) | | 1278 | 132 | | 2.44 | 24 | 700 ± 200 |

[*] The flux rate Q on these two methane plumes is computed considering half IME and half of the number of pixels due to the particular dual-plume structure caused by the angular configuration for that day (see Fig. 11).

Each flux rate $Q$ in Table 1 incorporates an attached uncertainty $k=1$ which is the result of the Monte Carlo propagation formalism described in subsection 2.3. These uncertainties cover a range of 15 to 45% which is a substantially lower figure compared to other missions such as PRISMA (Guanter et al. 2021). The main factor of disagreement resides in the different slope values of the $U_{10}$-$U_{eff}$ relationship applied to plumes with L lower than 200 m. The slope associated to PRISMA is 0.34 whereas the simulations for WV-3 result in a slope of just 0.12 (see Eq. (4)). The uncertainty of $U_{10}$ is, at the moment, the dominant uncertainty source and, thus, the lower slope coefficient for WV-3 contributes to the lower $Q$ uncertainty.

The fact that $U_{eff}$ (or $U_{10}$) represents the dominant source of uncertainty is studied in Figure 10. It shows the sorted $Q$ values in ascending order at the bottom panel with the associated relative uncertainty for each plume at the top panel. The figure captures an uncertainty trend associated to the $Q$ level with a small correlation of 0.45. The same ordering methodology has been replicated for the IME and $U_{eff}$ values. The results showed a weak correlation of the $Q$ uncertainty with the IME value (~0.15) and a high correlation (~0.85) of the Q uncertainty with the $U_{eff}$.



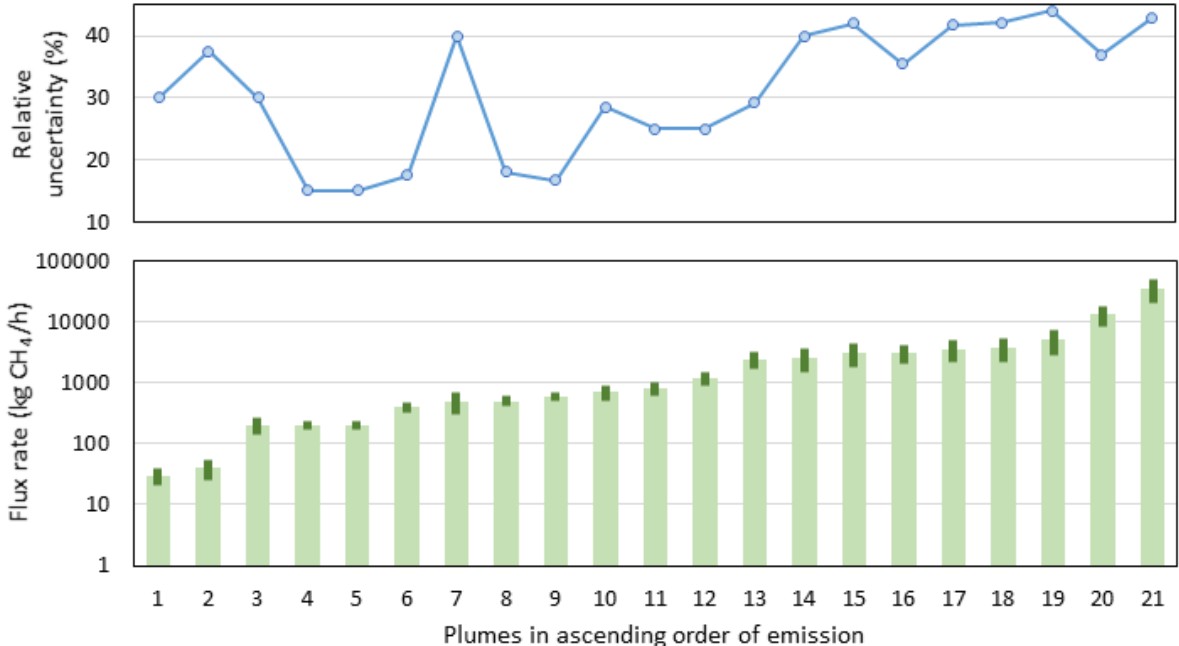

**Figure 10. Distribution of emission rates in ascending order for the 21 estimated Q flux rates listed in Table 1, which correspond to the 26 single methane plumes detected with WV-3. The bars at the bottom panel for each Q (the axis is in logarithmic scale) enclosed at its upper end the corresponding ±σ value displayed by thinner and darker bars. Top panel shows the relative uncertainty in % for each associated emission of the bottom panel.**

Figures 11-15 showcase the methane plumes listed in Table 1 as part of the retrieved methane enhancement maps of the studied areas. In some cases, the methane plumes have been cropped and the WV-3 SWIR B1 band has been integrated as a background scene for visual identification of the underlying infrastructure. Figure 11 displays the $\Delta XCH_4$ map from Turkmenistan South on 29[th] March 2021 at the left panel. This map includes two plumes that are zoomed in and displayed with the WV-3 SWIR B1 background. These two methane plumes present a particular shape as a consequence of the orthogonal alignment of the sun-to-satellite plane with respect to the wind direction (Figure 11 includes a schema of the angular configuration). This particular angular configuration enhances the parallax effect introduced by the plume at a certain height and is translated into two different plume projections (sun and surface reflected paths) that are clearly visible in the panel b) from Figure 11. This example is similar to the dual-plume structure described in Borchardt et al. (2021) for AVIRIS-NG and comes to reinforce the importance of geometric considerations for high-spatial methane quantification. Furthermore, the particular angle configuration in Figure 11 also enhances the effect of soil morphology visible throughout the entire $\Delta XCH_4$ map as a NW-SE streak noise pattern. Despite the geometric and soil morphology effects, both emission sources are easily distinguished by the high $\Delta XCH_4$ values and the unmistakable diffuse shape of the tail downwind. These two emitter points have been already registered by Irakulis-Loitxate et al. (2021b). The massive southern plume extends up to 3 km as shown in the enhancement map b) -white box in the left panel. However, the retrieved mask used to calculate the





IME and associated flux rate Q only takes the most integral part of the plume as the zoomed map in panel b) shows. Moreover, because the parallax effect, only half of each plume mask would be considered for flux rate Q estimates. This plume emerges from a gas pipeline near the compressor station already reported by Varon et al. (2019; 2021); and Irakulis-Loitxate et al. (2021b). Despite the first announcement in Varon.et al. (2019) and subsequent shutdown of the leak, it is here shown that this emitter source is once again spreading methane to the atmosphere. On the North area of the left panel in Fig. 11, another plume with a flux rate four times smaller is also detected. Both points are classified as O&G ground flare emitters that are often used in the burning of gaseous waste and whose installation is distinguishable on the images.

**Figure 11. ΔXCH₄ map produced from the real image of Turkmenistan South; Korpezhe O&G field on 2021/03/29 where two emissions are detected following a particular dual-plume structure consequence of the angular configuration on that day shown in the left diagram. Zoom panels on the right show the plumes over the band 1 of the © WV-3 SWIR image used as a background. The plumes in a) and b) comes from two different ground flares located respectively in coordinates 38.557º N, 54.200º E; and 38.494º N, 54.198º E.**



The area with the largest number of emitters in this study corresponds to the Goturdepe O&G field and is located in the North of Turkmenistan. A total of 16 different plumes are detected throughout this basin, one of the oldest oil piping systems in Turkmenistan. The top panel in Fig. 12 shows the methane enhancement map for the Eastern image on 10th April 2021. Two elongated plumes (consequence of strong wind; see Table 1) are highlighted in the image inside a white rectangular box. The plumes originate from two-point sources -registered also in Irakulis-Loitxate et al. (2021b)- where the dark color of the soil in the 2021 Google Earth RGBs images indicate that oil burning works are taking place in the area. The biggest plume (Fig. 12a) emerges from a thin pipeline coming out of the oil power plant adjoining. The origin of the other plume (Fig. 12b) is not so obvious but is likely caused by leakage from the piping system affecting the area.

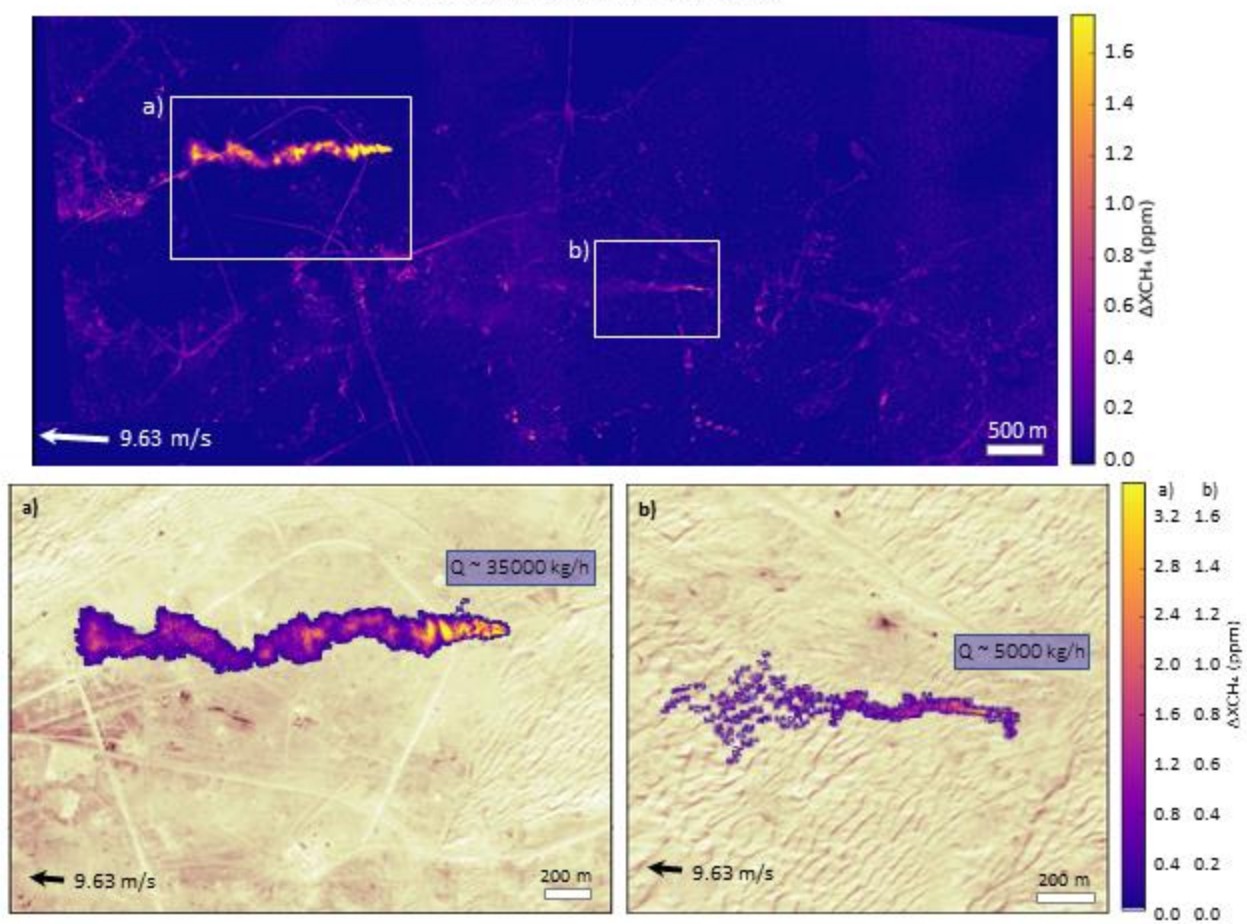

**Figure 12. ΔXCH₄ map produced from the real image of Turkmenistan North (Eastern image); Goturdepe field on 2021/04/10 where two emissions are detected. Zoom panels show the plumes over the band 1 of the © WV-3 SWIR image used as a background. The plumes in a) and b) comes from two different pipelines located respectively in coordinates 39.474º N, 53.743º E; and 39.462ºN, 53.775ºE.**



The image covering the latter area of the North of Turkmenistan on 29th March 2021 is perhaps the most interesting one because of the number of small methane leakages that WV-3 has been able to reveal (see Fig. 13b). Looking at a first glance at the ΔXCH4 map, an apparent single plume at the well-known point northwest of the image (Irakulis-Loitxate et al., 2021b)

390  can be glimpsed, as well as another plume towards the center of the image. However, by zooming in closer and considering the wind direction on that day, it has been possible to disentangle each of the underlying and independent emitting points for each case and even unfolding those principal sources. For example, observing this apparent first northwest plume near coordinates 39.498º N, 53.636º E it is found just a few meters away, a second emitter point at the northern end of the same pipeline where a ground flare is perceived. WV-3 data makes it possible to clearly attribute these two emitting points to

395  separate pieces of infrastructure, even if their tails intertwine later downwind. The same occurs with the plume seen towards the image center near coordinates 39.485° N, 53.663° E. This plume is in fact subdivided into four sources from different leakage points along an oil pipeline, shown zoomed-in in Fig. 14. Following its route to the south-east, we come across other methane plumes and quite a few smaller ones among the pipeline crossings to the south up to 50 m in length and a flux rate Q of about 30 kg/h.

400





**Figure 13. ΔXCH₄ maps from plumes detected in real WV-3 images acquired in three different parts of the world. a) Emitters from Hassi Messaoud Oil field in Algeria on 2021/01/17 (Pipeline: 31.742º N, 5.895º E), and on 2020/12/29 (Pipelines from left to right located in 31.778º N, 5.995º E; 31.768° N, 6.000° E; and 31.797°N, 6.011°E; detailed in Fig. 15). b) Emitters from Goturdepe O&G field in Turkmenistan North on 2021/03/29 with 14 plumes located along the pipelines: two-point source emitters NW of the image (39.498° N, 53.636° E and 39.497° N, 53.638° E); four plumes around 39.485°N, 53.663°E (detailed in Fig. 14); those further East zoomed around 39.480° N, 53.671° E; and the smaller southern in the surrounding of 39.469° N, 53.649° E. c) Emitters from Coal mines in Shanxi China on 2021/04/27 from left to right: Xiligaocun (36.2574° N, 112.9227° E); Taoyuancun (36.2337° N, 112.9458° E), Wangzhuang Beili two-point source emitters (36.2470° N, 112.9886° E and 36.2461° N, 112.9894° E). Band 1 of the © WV-3 SWIR images is used as a background.**

Some of the small leaks illustrated in Fig. 13b would go unnoticed by most satellites with lower spatial resolution and worse specifications for methane detection. Particularly those small plumes registered in the grey box area of roughly one square kilometer at the bottom right of the image.

Figure 14 exemplifies this fact by upsampling the real image from 3.7 m/pix to 30 m/pix and showing how the different point leakages would not be identified at coarser resolution and would be subsequently registered as a single plume instead of four independent sources. The ability to precisely locate and identify these emitting points could help steer the decision-making process for repairing such pipeline leaks, delivering strong environmental and socioeconomic impacts.

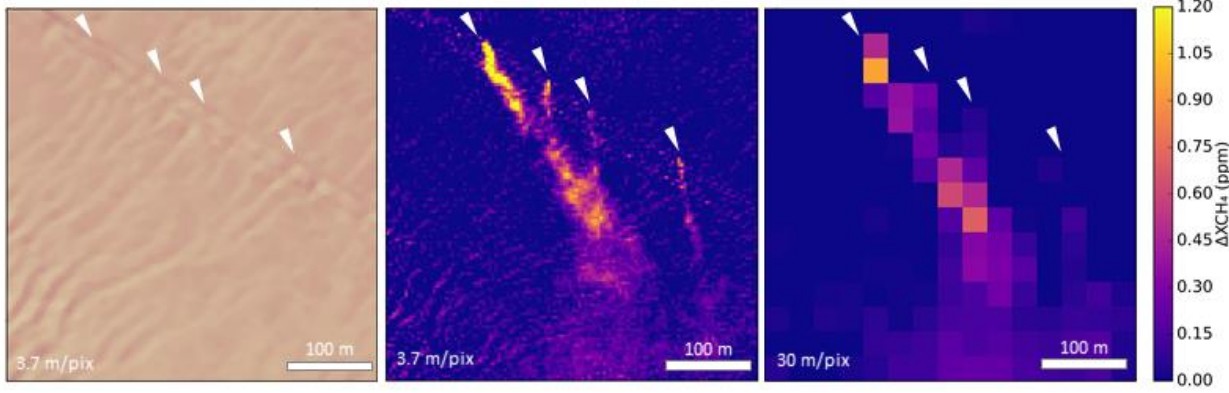

**Figure 14. Detailed example for Turkmenistan North; Goturdepe O&G field on 2021/03/29 where four plumes are seen coming out of a pipeline around 39.485° N, 53.663° E (zoom of Fig. 13b). Left panel uses Band 1 of the © WV-3 SWIR images as a background.**

Regarding the image in Algeria on the 29th of December 2020, despite optimal surface conditions, the results show a radiance jump around the middle of the image as it was already discussed in section 3.1. This is not the consequence of two different overpasses but from an unknown artifact (e.g. a staggered detector focal plane). In order to overcome this issue, the methane retrieval was processed separately for each part of the image. Figure 15 shows the resulting methane enhancement map where the visual results have clearly improved and just a residual effect at the fringes of the transition area. On this day three elongated plumes emerging from different pipelines are almost perfectly aligned with wind direction, and with emission concentration values steadily decreasing due to the relatively high wind speed (6.14 m/s). This high wind speed produces a fast dispersion of the plume at about 500 m travel, blurring until the plume tail disappears. On the contrary, on





days with low wind speeds (2.37 m/s) as it happens on the 17th of January 2021 (see example in Fig. 13a), the high concentrations of the gas remain concentrated near the emitting point defining a shorter plume.

To further demonstrate the value of methane plume detection at 3.7 m pixel resolution, Fig. 15 shows the ΔXCH₄ map obtained from the Algeria image on 29th December 2020 (see isolated plumes in Fig. 13a) and compares it with this same

result but upsampled to 30 m resolution similarly to Fig. 14. With this coarser resolution, apart from blurring the emitter point, the plumes become more difficult to define, especially the smaller one which is easily confused amid the background noise.

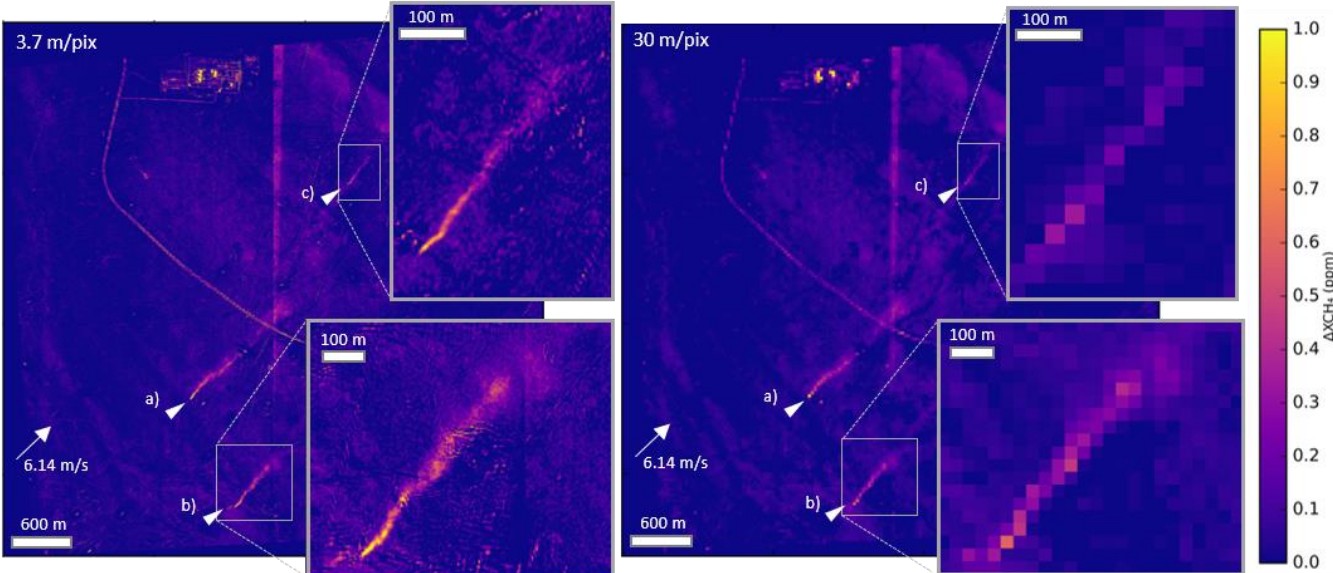

**Figure 15. ΔXCH₄ map for the image on 2020/12/29 over Algeria; Hassi Messaoud Oil field. Methane emissions from pipelines in**
**a) (31.778º N, 5.995º E), b) (31.768° N, 6.000° E) and c) (31.797°N, 6.011°E) locations; the latter two zoomed. Left panel correspond to the ΔXCH₄ map derived from the real WV-3 image (3.7 m/pixel) while the right panel shows the same map but resampled to 30 m resolution.**

Moreover, the high spatial resolution of WV-3 has not only been helpful to identify small sources of methane emissions but
also to detect outliers. Based on the idea that a methane plume in the atmosphere is characterized by soft edges whereas the presence of other features such as buildings and roads typically result in sharp edges, it has been possible to apply feature detection techniques as described in subsection 2.2.

## 4 Summary and outlook

This study reveals the previously undocumented capability of WV-3 for mapping methane point source emissions at very
high spatial resolution. The retrieval methodology is based on the MLR of 6 SWIR bands with low sensitivity to methane absorption against B7 (2235-2285 nm), which is positioned at a highly-sensitive methane absorption region. The ratio



between the original B7 band and the band obtained through the regression results in an estimated methane plume transmittance that links to the column-averaged mixing ratio ($\Delta$XCH$_4$). The study found a weak sensitivity of the methane retrieval to the SRF of B7 and a negligible effect to the atmospheric water vapor column. Although WV-3 B8 (2295-2365

nm) is at a stronger methane absorption position than the rest of the bands, our methodology proved to retrieve better results using B7 as a reference due to its higher spectral correlation with the rest of the SWIR bands.

The unprecedented spatial resolution of WV-3 combined with its high-quality spectral configuration and relatively low noise enables the detection of even smaller plumes than detected by hyperspectral missions (~30 m pixel resolution) and other multispectral missions such as S2 (20 m pixels). Furthermore, this mission includes unique pointing capabilities which can

optimize acquisition over O&G fields by tracking directly over suspected areas of point emitters with a daily revisit or better (E. S. Imaging, 2020).

The resulting $\Delta$XCH$_4$ maps obtained with our methane retrieval from WV-3 SWIR images have identified different uncontrolled gas leaks in two different O&G extraction regions in Turkmenistan and Algeria, which in some cases come directly from the production facilities and in others are due to ruptures of the natural gas pipelines, and some emissions in the

Shanxi coal mining region in China. The precise detection and identification of these emitters enables direct action to mitigate them. Our results show how we can pinpoint small emissions along oil pipelines (<100 kg/h) as shown in Fig. 13 and Fig. 14, illustrating the unique capability of WV-3 for methane mapping from space. WV-3 has superior detection limit to other multispectral satellite missions, comparable in some cases to airborne spectrometers.

The end-to-end sensitivity analysis of the methane retrieval with WV-3 has shown excellent results over homogeneous

surfaces. The IME loss for methane sources over an O&G extraction region in Algeria was as low as ~10 Kg (21% loss for a flux rate of 1000 kg/h). For highly heterogeneous areas such as the scene in Shanxi, China, larger fractions of the IME can be lost (up to 63%, or ~153 Kg for a flux rate of 1000 kg/h).

Despite the promising results shown in this first assessment of WV-3 for monitoring and quantifying methane plumes, further work is needed to diminish the impact of the surface background and the plume parallax effects on the derived

methane concentration enhancement maps. The former issue is shared with other missions, but in the case of WV-3 it becomes critical due to the higher variability of the surface at 3.7 m sampling and its orbit pointing capabilities that can result in large viewing zenith angles. The latter issue is enhanced at angular configurations such as the one presented in the example of Korpezhe O&G field in Fig. 11. In the case of the plume parallax effect, 3-D information about the plume structure is needed to estimate a path at a pixel-level that supports a more accurate quantification. The surface variability

could be mitigated if further development of the retrieval includes knowledge of the surface local angle information (e.g. through a high spatial resolution Digital Elevation Model). In addition to the SWIR bands, the WV-3 mission also images the VNIR TOA radiance with 1.24 m spatial resolution at nadir. A combination of the SWIR images with this VNIR data



could improve both the regression and the identification of emission infrastructure. Moreover, both the simulations and real cases investigated here are for an acquisition period between October 2020 and April 2021 characterized by a lower SZA

around the winter solstice in the Northern Hemisphere. Thus, it is expected that the potential of plume detection could increase if acquisitions around the summer solstice and Northern Hemisphere were considered (e.g., due to better SNR and lower shadowing).

In summary, the unique configuration of the WV-3 mission can substantially benefit current mapping efforts with a positive impact on the definition of a future methane observing system as proposed by recent EU regulations (COM, 2020). Very

high-resolution satellite observations of methane point sources as from WV-3 could play an integral role in future near-real time emission detection service.

## Acknowledgements

First, we would to thank the European Space Agency and European Space Imaging for the access to the WV-3 data through the Third-Party Missions scheme. We personally thank Silvester Fischer and Thierry Buettel from European Space Imaging

for their assistance in the product details and processing. We thank Daniel Cusworth (JPL) for the access to realistic methane plume simulations with WRF-LES. Javier Gorroño is funded by the ESA Living Planet Fellowship.

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
