# Peer review of "Mapping methane plumes at very high spatial resolution with the WorldView-3 satellite"

_Atmospheric Measurement Techniques, 2021_

## Author Comment (AC1)

Elena Sánchez García
Universitat Politècnica de València
Camí de Vera s/n,
46022, València

29th November 2021

Dear Editor,

Please find enclosed the revised manuscript entitled "Mapping methane plumes at very high spatial resolution with the WorldView-3 satellite" that we submitted for publication in Atmospheric Measurement Techniques, Manuscript ID: amt-2021-238.

In this new version we have taken into account the remarks and suggestions that the reviewers made, which were constructive and useful, and improved the quality and the understanding of the paper. The detailed explanation of the changes and corrections introduced in the new version is attached below. Other minor changes have been done to update the manuscript considering it was sent four months ago.

We kindly thank you for your interest and attention, and we hope that the revised version is satisfactory. If there are any further changes that you desire, then we are most willing to consider them.

Sincerely yours,

Elena Sánchez García
Researcher Scientist, Corresponding author

**DETAILED EXPLANATION OF CHANGES INTRODUCED**

*__Answer to Reviewer 1__* (Vladimir Savastiouk)

Suggestions for minor edits
We thank the referee for his beneficial suggestions. All of the minor changes made following the referee's suggestions are as follows:

**Comment 1:**
Referee: 1. I am not an expert on methane retrievals, but while the paper has a good discussion about the uncertainties it does not give clear guidance on the absolute level of methane concentrations that can be detected with this technique. It will be useful to know, for example, whether this method can be used for detecting permafrost thaw methane leakage in the Arctic.
Answer:
The main discussion about the potential levels of methane detection can be found in subsection 3.1 and the main findings are illustrated in Figure 9. This graph presents the IME fraction that can be detected for different sites and the minimum flux (detection threshold) achievable for those simulated sites. In addition, with the real cases presented in the manuscript, it offers a first assessment of the potential of WV-3 for point-source methane emissions.
In the reviewed version we have upgraded Figure 9 with a refined sampling of flux rates for a better understanding of the detection threshold at the three sites. We have also clarified in the text the flux rate levels that can be detected and their relationship with the real cases shown.

Further research is needed to understand its application to other sites/emission types. The example of the permafrost suggested by the reviewer would be very interesting but challenging due to the high latitude (large SZA and heterogeneity) and the diffuse nature of the emissions. Because of its high SNR, methane mapping in water bodies (e.g. off-shore wells) could be an interesting area of study with WV-3.

**Comment 2:**
Referee: AMF calculations require some knowledge of the vertical distribution of the absorber. There is no indication in the paper as to what distribution is used for methane plumes.
Answer:
The AMF is calculated as: AMF = $(\cos^{-1}(SZA) + \cos^{-1}(VZA))$. This calculation assumes a plane parallel purely absorbing atmosphere.
We have updated the term in the manuscript as "geometric air mass factor" to specify this point.

**Comment 3:**
Referee: The claim that the usage of Time-Delayed-Integration (TDI) of 16 lines contributes to a superior SNR is not supported by a reference or by an explanation of why this is the case. It may be trivial to the authors, but maybe of interest to some non-expert readers.
Answer: TDI refers to the exposure of the same area multiple times by different pixels in the detector as the satellite is moving. These measurements can be added and consequently improve the noise of the image. In an ideal scenario under a perfect registration, the noise would improve by the square root of the TDI stages. WV-3 design with narrow spectral bands and large spatial resolution means that the energy collected at a pixel-level is a-priori small. However, the products delivered by WV-3 contain 16 TDI stages which contribute to largely reducing the noise (ideally noise improves by 4 from a single acquisition).

The revised manuscript includes an improved explanation of the TDI and its effect on the SNR. Regarding the SNR study, we thought not to focus so much so as not to distort the main objective of the paper. We explain here in more detail:

The study of the WV-3 instrument noise is based on the SNR estimation over one of the selected sites for simulation (see Algeria site in Fig. 5). The dataset for WV-3 is a tasked acquisition at approximately 24.8° VZA and 57.2° SZA over the Algeria O&G production facilities on the 2020/12/29 centered. The considered dataset for S2 is *S2A_MSIL1C_20201224T101431_N0209_R022_T31SGR_20201224T121510*. This is just a few days apart since the acquisition on the same day included cirrus over the selected area. The selected area for both missions is approx. $1.7 \times 1.7 \text{ km}^2$ for both missions at latitude 31.7791° N and longitude 5.9884° E.

The method to estimate the SNR calculates the variance of the higher-order coefficient of the Discrete Cosine Transform (DCT) similarly to (Alonso et al., 2019). The obtained S2 B12 SNR is approximately 290 while the SNR for WV-3 is approximately 400 and 430 for B7 and B8 respectively. Although the optical design of WV-3 is based on narrow bands and larger spatial resolution pixels, the instrument noise is excellent due to the processing of the images in Time-Delayed-Integration (TDI) of 16 lines as compared to 2 lines for the S2 mission.

---

## Author Comment (AC2)

**DETAILED EXPLANATION OF CHANGES INTRODUCED**

*Answer to Reviewer 2* (Folkert Boersma)

Suggestions for improvements and minor revisions.
We thank the referee for his words and thoughts about our work. All of the minor changes made following the referee's suggestions are as follows:

**Comment 1:**
Referee: The retrieval technique appears straightforward and uses relatively broad spectral bands in which CH4 has a differential absorption strength. The technique is aimed at detecting enhancements rather than absolute values, and therefore compares the depth of an 'on' (strong absorption) CH4 band to a spectrally nearby 'off' (weak absorption) band, and explains the difference by the integrated amount of CH4 along the photon path.
The implicit assumption is that there are no other processes that result in spectrally varying signals at the sensor (surface emissivity, other absorbers, instrument issues) for the two bands. This is evident from Eq. (2) which presupposes the existence of a direct relation between the methane enhancement and the 'on/off' ratio of radiance signals. It furthermore remains unclear how the air mass factor (AMF) has been calculated, and whether the AMF may be assumed to be spectrally constant. Also, it remains unclear how surface heterogeneity would influence the CH4 retrieval, or how surface effects could be accounted for. I think the retrieval approach should be discussed in much more detail, providing more justification for why certain steps are taken or why simplifications have been made.
Answer:
We thank the referee for his constructive comment.
The detection and quantification of methane point-source emissions from space is a novel area and the integration of multispectral mission is even more recent. Nonetheless, we have proposed a methodology, proved it against both simulated and real cases, and assessed the associated uncertainty. Section 3.1 provides a more in-depth study of the retrieval limitations at three different sites with very different surface conditions.
The AMF refers to the geometric air mass factor as explained in detail in the minor comments below. It is applied at B7 (~2200nm).
Surface heterogeneity is indeed one of the major limitations for methane mapping from space. It is a big challenge for hyperspectral missions and even more challenging for multispectral ones. One potential improvement that we point to in the manuscript is the possibility to correct topographic effects. However, these and other corrections are still under development.

**Comment 2:**
Referee: Then the paper lacks a discussion on the WV3 satellite specifics, such as orbit, overpass time, spectral coverage, and spatial coverage. This makes it very difficult to judge what is now the true potential of WV3 to detect plumes around the globe, and with what revisit time, taking into account cloudiness. The paper also would benefit strongly from or perhaps even need an evaluation of the CH4 enhancements against independent CH4 data, such as for example from Sentinel-2 or TROPOMI. Without such an evaluation, it remains difficult to properly judge the potential of the interesting CH4 enhancements to serve as a quantitative source of information rather than just images of possible CH4 plumes.
Answer:
WV-3 is a multispectral mission with pointing capabilities and spatial resolution of 3.7 m in the SWIR. Its pointing capabilities mean that it can acquire any predefined location in less than a day. We have demonstrated that its unique spectral and noise design, can map methane emission with great accuracy and low detection threshold.

The true potential of WV-3 can be applied to the monitoring of industrial emissions such as O&G fields or other critical infrastructure (e.g. oil pipelines) where an almost daily acquisition over a predefined area would be possible. Its spatial resolution is able to precisely pinpoint the location of the leakages and its low detection threshold would be helpful to detect most of the methane releases.

Concepts such as its main application and the revisit time are included in the manuscript. However, we have considered to highlight that in the reviewed version by including the revisit time in the abstract to clarify this point.

The detection and quantification of point-source emissions with multispectral satellites is a very recent field of study. The first demonstration of the capabilities of S2 to detect methane point source emissions was recently published in Varon 2020. The validation against other missions is ongoing but most of the retrieval methodologies are still under refinement.

Its comparison against TROPOMI is also challenging due to the large differences in spatial resolution between the missions.

The approach developed here for validation simulates methane products using WRF-LES plume emissions. These products constitute our benchmark against which we can test the retrieval. For example, a similar approach has been developed with S2 preliminary validation products available in: https://doi.org/10.7910/DVN/KRNPEH.

In this reviewed version, section 3.1 provides a more in-depth study of the retrieval limitations at three different sites with very different surface conditions. The simulation has been thoroughly reviewed and Figures 5-7 have been changed. In this new version, we found that the original plumes were oversized and the zoom ratio has been corrected. This has a minor effect on the validation results in Figure 9. Indeed, Figure 9 has been improved with a fine flux rate sampling to better reflect the impact of different surface conditions in the detection threshold.

**Minor comments:**

Referee:

P1, L8-9: detection of methane alone is in itself not a mitigation strategy, although it can be the start of one.

Answer:

Following the referee's advice, we have changed the sentence to sound less forceful. So, we change "The detection of methane emissions from industrial activities has been identified as an effective climate change mitigation strategy" to "The detection of methane emissions from industrial activities is a cornerstone for the preparedness of effective climate change mitigation strategies".

Referee:

P3-4: please clarify what "WV-3 images are processed with a Time-Delayed Integration of 16 lines means". I guess this has to do with co-adding spectra, but it is not entirely clear. What is the ultimate signal-to-noise level after co-addition?

Answer:

TDI refers to the exposure of the same area multiple times by different pixels in the detector as the satellite is moving. These measurements can be added and consequently improve the noise of the image. In an ideal scenario under a perfect registration, the noise would improve by the square root of the TDI stages. WV-3 design with narrow spectral bands and large spatial resolution means that the energy collected at a pixel-level is a-priori small. However, the products delivered by WV-3 contain 16 TDI stages which contribute to largely reducing the noise (ideally noise improves by 4 from a single acquisition).

The revised manuscript includes an improved explanation of the TDI and its effect on the SNR.

Referee:
P4, Eq. (1): please define the air mass factor in this case. How is it calculated?
Answer:
The AMF is calculated as: AMF = (cos−1 (SZA) + cos−1 (VZA)). This calculation assumes a plane parallel purely absorbing atmosphere. We have updated the term in the manuscript as "geometric air mass factor" to specify this point.

Referee:
L101: band 5 still has CH4 absorption …
Answer:
Thanks for pointing at this. We have updated the text accordingly for a better understanding in the 2.1 section when we describe the retrieval method by adding B5 in the explanation. The new paragraph looks like this:
"From the several options that have been considered (see Fig. 6), the selected method for the estimation of this "methane-free" band is based on a multiple linear regression (MLR) of B1-B6 with B7 as the target band (B7/B7$_{MLR}$ method). Despite B8 being more sensitive to methane absorption, this band has shown a lower spectral correlation with B1-B6 bands as compared to B7 (see results in subsection 3.1). Out of the six bands considered for the regression, three of them (B3, B5, and B6) are marginally sensitive to methane. In the case of B3, its weight on the regression is small and has negligible impact thus it has been decided not to include this band. However, in the case of B5 and B6, its spectral closeness to B7 improves the regression and outweighs the impact that the residual methane sensitivity of these bands produces on the retrieval."
Here we attach a more visual explanation of this residual absorption in B5 and B6.

[Figure]

As we pointed in the manuscript (particularly in section 3.1), we estimated a loss of around 7% in the IME. This test simulated the retrieval with an added methane plume on B7 and no effect on the rest of the bands (including B5 and B6). Further iterations of the methodology might seek a correction procedure for this effect.

Referee:
L104: why does band 8 show less spectral correlation with bands 1 to 6? Can there be saturation of the CH4 signal in band 8?
Answer:
This question is related to the previous one. B8 covers a spectral range that is more sensitive to methane absorption than B7. However, this same band is also at a larger spectral distance to B1-B6 and consequently, it exhibits less spectral correlation. This implies that an estimation of a band with no excess methane present becomes more difficult for B8 rather than B7.

Referee:
P4: please specify what the integrated mass enhancement is. I guess it is all the CH4 higher than background levels within the contours of the plume, but this may not be immediately clear to everyone.

Answer:
We appreciate the referee's comment because it is true that we clarified this term in line 150 but this explanation should appear the first time we talk about the IME. Then, we change this sentence that was later in line 150 "which refers to the measure of the total excess mass of observed methane widely explained in Frankenberg et al. (2016) and Varon et al. (2018)" to P4 (line 109-110).

Referee:
L129: awkward to state that retrievals "are confused".

Answer:
It has been changed "confuse" by "alter" for a better understanding.

Referee:
L131: here and later please specify at which altitude you take the windspeed to drive the plume, and why.

Answer:
In line 176 to explain about the wind speed considered we write: "the measurable 1-h average 10-m wind speed (U10) derives from the two north-south and east-west wind components of GEOS-FP dataset (https://portal.nccs.nasa.gov/datashare/gmao/geos-fp/das/) at the satellite acquisition time and for the location of each plume". Consider that GEOS-FP is a meteorological reanalysis product with a resolution of $0.25° × 0.3125°$ (Molod et al., 2012)".
Then U10 refers to the wind speed under neutral conditions averaged over one hour at 10 m above the sea surface. However, we agree with the referee that in line 131, which is the first time we talk about wind speed data, a small clarification as "10-m wind speed and direction data" would be anyway helpful so it has been added.

Referee:
L144-146: sentence starting "Techniques such as … in the retrieval" is difficult to follow. Please rephrase or clarify.

Answer:
We thank the referee for his suggestion and we change the sentence for a better understanding. We rephrase to "Feature recognition automatically detects the plume shape and removes features that, despite its high $\Delta$XCH4 values, are not part of the emission".

Referee:
L153-154: how can you assume that all the CH4 is in an 8 km column?

Answer:
This is just a conventionalism that has been used to date in all available literature considering that 8 km is the smallest thickness of the troposphere reached at the Earth's poles.

Referee:
L157: please explain what the effective wind speed is. Is this the wind speed profile weighted with some sort of CH4 number density profile? Figure 3: why is it that the effective wind is so much lower than the 10-meter altitude wind?

Answer:
The effective wind speed should be thought of as an operational parameter that maps a detected IME to a source rate Q, given a length scale L. Varon et al. (2018) is a very good reference for the understanding of this term. This is not a measurable quantity but can be

related (for example) to the measurable 10-m wind. The definition of L affects the shape of the Ueff = f(U10) relationship. The reason we need such low Ueff values here (much lower than U10) is because the L values (area of plume mask) are very low for small, high-res plumes. The plumes here are quite small/thin, so the sqrt(area) metric produces lower length scales than you would get from a plume-length metric.

Referee:
Figure 4: we are seeing a WRF LES-simulated plume but for which altitude are the mixing ratios depicted?
Answer:
Figure 4 is shown a column-integrated plume, so this is showing the total vertical column mixing ratio.

Referee:
L211: is the spectral optical depth defined as the vertical (rather than slant) optical depth? It seems to me that the viewing geometry (AMF) needs to be accounted for.
Answer:
The AMF is accounted for but we did not included that in the description. Thanks to the reviewer for the comment. We included the following sentence: "The optical depth is weighted by the AMF that considers the slant optical path due to both the illumination and viewing angular conditions."

Referee:
L217: what is the source of information for the "TOA radiance scene"? It seems to be the observed radiance, but for which scenes, viewing geometries, surface properties then?
Answer:
We have clarified the sentence in the manuscript.
The convolved plume transmittance by the WV-3 bands is multiplied by the observed radiance at each one of the bands (mentioned as "TOA radiance scene"). The scene, surface properties and geometries are exactly the ones of the satellite product bringing realistic scenarios.
Because the plume transmittance is multiplied directly to the observed radiance, this produces a bias. We try to compensate this error by introducing a correction that models this bias as follows:

$$\Delta L_{TOA} = \frac{\int T_{tot}(\lambda)T(\lambda)d\lambda}{\int T_{tot}(\lambda)d\lambda \int T(\lambda)d\lambda}$$

where $T_{tot}$ represents the transmission of all gases in the atmosphere withouth considering the methane plume and $T(\lambda)$ represents the plume transmittance.

Referee:
In Fig. 15 there are some straight lines appearing as CH4 enhancements along the north-south direction. What explains those lines?
Answer:
We thank again the referee for his careful review. This fact is mentioned around lines 264 in section 3.1 with the simulated Algeria's product.
Different "collection scenarios" are considered in the VW-3 mission that include the generation of an image with several stripes of different overpassess with a swath of 13.1 km. We explained in the paper that this effect in the Algeria WV3 image is caused during the WV-3 product processing and could be due to focal plane discontinuities or other acquisition artefacts which creates two different halves in the original image. However, in the derived methane maps, this effect must be considered a minor one largely due to the spectral correlation nature of the error.

Nonetheless, in the results (subsection 3.2), we compensate this effect by processing separately each half of the original image to obtain the methane map. This is why in Figure 5 these differences have been compensated, except for these intermediate lines of union between both halves to which the referee refers that remain.